# Transcriptomic and Proteomic Analysis Reveals Mechanisms of Patulin-Induced Cell Toxicity in Human Embryonic Kidney Cells

**DOI:** 10.3390/toxins12110681

**Published:** 2020-10-29

**Authors:** Nianfa Han, Ruilin Luo, Jiayu Liu, Tianmin Guo, Jiayu Feng, Xiaoli Peng

**Affiliations:** 1Beijing Jihua Biotechnology Services Co., LTD, Beijing 102200, China; nianfa.han@gene-van.com; 2College of Food Science and Engineering, Northwest A&F University, Yangling 712100, China; lrl545@nwafu.edu.cn (R.L.); liujiayu1002@nwafu.edu.cn (J.L.); guotianmin666@nwafu.edu.cn (T.G.); fengjiayu@nwafu.edu.cn (J.F.)

**Keywords:** patulin, renal cytotoxicity, HEK293 cells, transcriptomic, proteomic, apoptosis

## Abstract

Patulin (PAT) is a natural mycotoxin that commonly contaminates fruits and fruit-based products. Previous work indicated that PAT-induced apoptosis in which reactive oxygen species (ROS) are involved in human embryonic kidney (HEK293) cells. To uncover novel aspects of the possible mechanism of PAT nephrotoxicity, the transcriptome and proteome profiles were investigated using the digital gene expression (DGE) and isobaric tags for relative and absolute quantitation (iTRAQ) proteomic approaches. A total of 127 genes and 85 proteins were found to express differentially in response to 5 μM PAT for 10 h in HEK293 cells. The most dramatic changes of expression were noticed with genes or proteins related to apoptosis, oxidative phosphorylation ribosome and cell cycle. Especially, the activation of caspase 3, UQCR11, active transport form and endocytosis appeared to be crucial in PAT kidney cytotoxicity. PAT also seemed to be associated with cancer and neuropathic disease as pathways associated with carcinogenesis, Alzheimer’s disease and Parkinson’s disease were induced. Overall, this study served to uncover overall insights associated with signaling pathway that modulated the PAT toxicity mechanism.

## 1. Introduction

Patulin (PAT) is a highly toxic secondary metabolite produced by *Aspergillus* sp. and *Penicillium* sp., it can be commonly found in rotten fruits and crops as well as their derivative products [1,2]. At present, controlling PAT contamination remains a huge challenge. The accumulation of PAT presents a food safely risk and health hazard to human. PAT can cause edema, hemorrhage, immune dysfunction and exhibits nephrotoxic, hepatoxic, carcinogenic, neurotoxic and genotoxic properties [3,4]. 

A variety of mechanisms have been conducted for how PAT works, including forming covalent adducts with essential sulfhydryl groups [5], leading to mitochondrial dysfunction and endoplasmic reticulum stress [6], and the induction of excessive reactive oxygen species (ROS), which induce oxidative DNA damage and membrane peroxidation [7,8,9]. Moreover, mitogen-activated protein kinase and P53, Bcl2 family members as well as autophagy were also reported involved in PAT toxicology [10,11]. 

Understanding the precise toxin mechanism at molecular level is essential to predict potential deleterious effects and to develop suitable countermeasures to improve human health. To obtain a comprehensive view of the mechanism of PAT nephrotoxicity, here we simultaneously used transcriptomic and proteomic approaches to clarify the complex pathological changes in human embryonic kidney cells. It was found that (i) a number of pro-apoptotic genes and positive regulation of apoptosis proteins were up-regulated; (ii) complex III of the mitochondrial respiratory chain gene was enhanced, while ATP synthase was inhibited and complex I was disordered; and (iii) ribosome, cell cycle, nucleotide metabolism, cell growth, and endocytosis, as well as some disease-associated genes and proteins, played a relatively large role in PAT toxicology. 

## 2. Results

### 2.1. PAT Triggers Caspase-Dependent Cell Death via the Intrinsic Apoptotic Pathway

Previous studies using HEK293 cells have shown PAT-induced apoptosis and the collapse of mitochondrial membrane potential. We further confirmed the inhibition of cell viability using WST methods (Figure 1a,b). Following treatment with PAT, cells showed a significant decrease in cellular ATP content (Figure 1c,d). PAT also induced the release of Cyt c from mitochondria to cytosol preceding cell death (Figure 1e,f). Furthermore, activations of the apoptotic biomarkers, caspase cascades, were also detected. A marked elevation of activities of initiator caspase 9 and the downstream executioner caspase 3/7 was observed (Figure 2a,b). Co-treatment of caspase 3 specific inhibitor Z-DEVD-FMK protected cells from PAT-triggered cell death (Figure 2c) and LDH release (Figure 2d), and almost completely reversed the increase of enzyme activities of capase 9 and 3/7 at 10 h (Figure 2e,f). Thus, PAT was able to trigger caspase-dependent cell death via the intrinsic apoptosis pathway. 

### 2.2. Transcriptome Analysis 

The differentially expressed gene profile was quantified using digital gene expression (DGE) analysis. Samples for DGE analysis were collected 10 h after incubation with or without PAT. The saturation analysis can be performed to check whether the number of detected genes keep increasing when the sequencing amount increases. The trends of saturation are shown in Appendix A and the statistics of the DGE tags are shown in Appendix A. After removing the low-quality sequences and adapter sequences, 3,282,379 and 3,518,370 clean tags were obtained in the control and PAT group, respectively. Heterogeneity and redundancy are two significant characteristics of mRNA expression. Certain types of mRNA have very high abundance, while the majority remain at very low levels of expression. The distribution of clean tag expression can be used to evaluate the normality of the DGE data. In the present study, the distribution of distinct clean tag copy number showed extremely similar tendencies. Among the distinct clean tags in the two sample libraries, only 3.23–3.33% possessed more than 100 copies. The majority of distinct clean tags (59.80–60.16%) had two to five copies, which indicated that the DGE data among the two libraries was normally distributed.

The Homo sapiens reference genome contains 22,748 genes and 94.49% of these genes have CATG sites. Finally, the tag mapping onto the Homo sapiens genome generated 13,016 tag-mapped genes for control group, 13,268 for PAT group. In detail, 127 genes were regulated at least twofold. Of these genes, 84 showed increased, while 43 showed decreased expressions (Figure 3a), indicating that a relative extensive transcriptional reprogramming occurred. The detail differential expressed genes were presented in Appendix A.

The Gene Ontology (GO) annotations of cellular component, molecular function and biological processes of the differential expression genes are presented in Figure 3b–d. Kyoto Encyclopedia of Genes and Genomes (KEGG) pathway enrichment analyses were enriched at Appendix A. In order to validate the quality of the DGE, six randomly selected genes were measured by real-time PCR. Quantification of the signals showed that except FIS1, the expression patterns of all other genes were in accordance with the DGE results, although the ratios varied to some extent (Figure 4a–g). 

### 2.3. Proteomic Analysis 

To obtain more information about the nature of PAT-induced toxicity, an isobaric tags for relative and absolute quantitation (iTRAQ)-based approach was pursued to quantify the global protein expression profiles. Changes in the abundance of proteins were measured and compared at two biological replicates. Eighty-five proteins were expressed differentially at the level of 1.2-fold, of which 50 were upregulated and 35 were downregulated (Figure 5a). The details of differential expressed proteins are presented in Appendix A. 

The GO annotations of cellular components were classified into 11 different categories, of which 33, 23, 22 and 9 proteins were found to be differentially expressed in the cell nucleus (38.8%), cytoplasm (27.1%), mitochondria (25.9%) and plasma membrane (10.6%) (Figure 5b–d). Proteins located in such organelles were necessary and pivotal in response to stress stimuli, providing defense against mycotoxins. At the functional level, most are involved in binding and enzyme activity. The enzymes appeared were most associated with hydrolase (14.1%), transcription cofactor (9.4%), transferase (9.4%) and oxidoreductase (8.2%). The blast annotations of biological processes revealed that the proteins were involved most in included cellular metabolism (58.8%), biosynthetic (23.5%), cell development (21.2%), response to stress (20%), signal transmission (15.3%) and cell communication (15.3%). KEGG pathway enrichment analysis resulted in 93 pathways, among which “apoptosis”, “oxidative phosphorylation”, “nucleic acid metabolism”, “amino acid metabolism” and “disease-associated” genes were enriched (Appendix A).

### 2.4. Integrated Transcriptome and Proteome Analysis 

The alterations of mRNA levels do not always result in similar alterations in protein levels and enzyme activities. Nevertheless, a combination of mRNA and protein expression pattern is required to fully understand the toxicity machinery. Hence, the protein–protein interaction networks of the altered gene and protein expression are shown in Figure 6. The presentative networks were profiled according to GO biological processes, molecular functions, or KEGG pathways and were integrated with the String database. In general, sustained PAT pressure exerted influences on apoptosis, oxidative phosphorylation, ribosome, cell cycle, nucleotide metabolism, cell growth, endocytosis and disease-associated pathways.

DGE analysis indicated that three detected proapoptotic genes PSENEN (4.26), DENND4B (2.24) and SLC7A5 (2.06). Among the seven differentially expressed antiapoptosis genes, four of them, HMOX1 (3.2), BCL2L21 (3.85), RNF34 (2.58), CIAPIN (2.68) were upregulated, while the other three, AATF (0.48), RHOC (0.46) and CDKN2D (0.42) were dramatically downregulated. In iTRAQ data, 10 in 11 of the apoptosis-related proteins, such as the positive regulation of apoptosis proteins J JUN (2.29), BCL10 (1.48), BCL7C (1.48), CCNA2 (1.48), CASP3 (1.47), PDCD4 (1.24), were upregulated.

A considerable fraction of the differentially regulated genes and proteins were associated with oxidative phosphorylation. The gene expressions of complex III (UQCR11, 2.47) and complex I (NDUFA4, 2.45) were induced by PAT exposure, while the gene expression of complex V (ATP6V1C2, 0.34), and all the five detected differentially regulated proteins, ATP6 (0.797), ATP5O (0.793), NDUFAF4 (0.737) and NDUFA5 (0.688), NDUFA6 (0.647) were significantly inhibited. 

The ribosome and associated molecules are also known as the protein translational apparatus. Five DGEs (RPS9, CSTF3-AS1, LINC01623, RPS20, RPLP2) encoded ribosome proteins were induced. However, RPL11 mRNA expression and all ten differentially detected expression proteins in iTRAQ data were inhibited significantly. 

All seven proteins and five genes involved in cell cycle were upregulated by PAT, while three genes involved in cell cycle were downregulated. Among these proteins or genes, RNA binding protein RBM38 plays a crucial role in cell cycle arrest via its binding to p21. The transcription factor FOXO4 has been shown to cause cell cycle arrest between the G0 and S phases. The widespread cyclin Cyclin-A2 (CCNA2) exhibits a promotion function both cell cycle G1/S and G2/M transition. CCKN2D (0.46) has been shown to function as cell growth inhibitor that controls cell cycle G1 progression. PKMYT1 (0.45) plays a role in the negatively regulation of cell cycle G2/M transition. The regulation of these genes and proteins suggested that PAT disrupted the cell cycle, which might further affect the final destiny of cells under PAT stress.

A considerable fraction of the regulated proteins and genes were associated with nucleotide metabolism. Two genes and four proteins related to RNA biosynthetic process were upregulated, while four genes and three proteins were downregulated. On the other hand, most of the differentially expressed genes and proteins related RNA degradation (three in four) as well as DNA damage and repair (six in seven) were increased. Moreover, two RNA transport-related genes, TBC1D17 (0.46) and POP7 (0.50) were decreased by PAT exposure.

Downregulation of three transcripts of the cell growth, including CDKN2D (0.46), RHOC (0.42), and CAPRIN2 (0.32), was observed. Two senescence-related genes, DBAJA3 (2.11) and HSD17B10 (20.8), were upregulated. Notably, another cell growth gene, KAZALD1, was significantly induced. Such results generally suggested PAT induced the disorder of cell growth and promoted the aging of the cells.

Four endocytosis-related genes, ULKL (3.02), GUCY1B3 (2.67), UIMC1 (2.44) and SCARB1 (2.34) were overexpressed at the mRNA level. Finally, three upregulated proteins, PIP5K1A (1.45), PARD6B (1.32) and PPP1RSP3 (1.36) were involved in endocytosis. 

In addition, several of the differentially expressed genes and proteins were functionally disease-associated, especially associated with carcinogenesis, Alzheimer’ disease and Parkinson’s disease. These findings provide support that PAT also seemed to be associated with cancer and neuropathic disease. 

## 3. Discussion

Here, in an attempt to uncover novel possible mechanism of PAT toxicity, exhaustive transcriptome and proteome profiles were investigated. In our research, only a few genes were altered simultaneously. The poor correlations between mRNA and protein large data sets were also reported by some comparative studies [12,13,14]. The possible explanations including: (1) protein abundance is regulated positively or negatively at the post-transcriptional, translational, and post-translational levels; (2) the half-life of mRNA and/or protein is quite different; (3) the intracellular location could also affected the extraction efficiency of proteins; (4) although 94.5% of the genes contain CATG sites could be recognized and cut off by NlaIII enzyme in DGE analysis, there is still part of the genes couldn′t be detected. As DGE and iTRAQ have their own limitations, a combination of mRNA and protein expression patterns are required to fully understand the mechanism of PAT toxicity.

While the precise mechanism of PAT’s toxicity remain to be fully elucidated, some evidences suggested that a range of epigenetic influences involving oxidative stress, glutathione modulation and apoptosis [8,10,15,16,17]. Our data indicated that apoptosis is a major mechanism of PAT induced cell death. We also demonstrated that PAT was able to trigger apoptosis by disruption mitochondrial function, inhibiting the production of energy, releasing cyt c to cytosol, ultimately initiating the caspase signal pathway.

PAT treatment led changes in proteins associated with apoptosis. Three proapoptotic genes and four antiapoptosis genes were upregulated. Most of the positive regulation of apoptosis proteins including CASP3 was upregulated. Besides, three antiapoptosis genes and one apoptosis protein were downregulated. Apoptosis is important for tissue homeostasis, but too much or too little apoptosis can also cause diverse effects [18]. In a cell, the balance between anti- and proapoptotic genes or proteins determines if a cell undergoes apoptosis or survives [19]. The upregulation of apoptosis executioner caspase 3, which was in line with the elevated enzyme activities, indicated the final destination of the cells under PAT stress. In fact, the activation caspase 3 has been reported in human leukemia cells [20], colorectal cancer cells [6,21] and in several organs in experimental animals [8,22,23] under PAT hazards.

Our previous study also provided evidence that PAT induced overproduction of ROS [9]. ROS scavenger, GSH, blocked the cell apoptosis evoked by PAT [24]. Excessive ROS destroyed the mitochondrial membrane integrity, yet, in another aspect, mitochondria have been deemed as one of the major sites producing ROS. In the process of ATP formation, 1–5% of the electrons will be leaked out to form an electron transport chain and to produce ROS [25]. At present, most research results confirmed that complex III is the main source of ROS, followed by complex I, whereas complex II can also generate ROS slightly. In our system, the gene expressions of complex III (UQCR11) and complex I (NDUFA4) were induced, while the protein expression of complex I (NDUFAF4, NDUFA5 and NDUFA6) were significantly inhibited. Complex III catalyzes electron transport form QH_2_ to cyt c, and 70–80% of mitochondrial superoxide radicals arise from the QH_2_ to ubiquinone cycle [26,27,28]. Our results implied that the dysfunction of electron transport chain, especially the elevated expression of UQCR11 might be able to explain the increase of ROS. However, this requires further verification. Beyond that, Nicotinamide adenine dinucleotide phosphate (NADPH) oxidase is another site of intercellular ROS production. 

Maintaining a normal state of the cells requires sufficient energy. Mitochondria provide more than 15 times the ATP produced by anaerobic glycolysis [29]. In our system, the gene expression of ATP synthase (ATP6V1C2), and two differentially regulated ATP synthase proteins (ATP6, ATP5O) were significantly inhibited. Besides, the accumulation of ROS may lead to irreversible oxidative damage of mitochondrial membrane lipids and proteins, as well as reduce the mitochondrial transmembrane potential. Loss of transmembrane potential can perturb the production of ATP, due to the fact that an electrochemical gradient is needed to provide the driving force for ATP synthesis. 

PAT is a low-molecular-weight, highly polar and hydrophilic molecule, but the phospholipid bilayer plasma membrane as well as the organelle membrane is hydrophobic. The improved expression of endocytosis related genes and proteins indicated that this active transport form, endocytosis, might be a general pathway for PAT to enter into the cells, thus triggering the cytotoxic effects, and finally inducing cell death.

## 4. Conclusions

This study presents evidence that PAT triggers caspase-dependent cell death via the intrinsic apoptotic pathway. A comprehensive transcriptome and proteomics analyses that combined two unbiased and high throughput techniques for the first time were used to define the mechanism of nephrotoxicity of PAT. Genes and proteins related to apoptosis, oxidative phosphorylation ribosome, cell cycle, nucleotide metabolism, cell growth and endocytosis were differentially expressed in HEK293 cells induced by PAT. Especially, caspase 3 appeared to be particularly crucial in PAT-induced cell apoptosis and the elevated expression of UQCR11 might be able to explain the increase of ROS. In addition, the activation active transport form and endocytosis, might be a general way for PAT to entry into the cells. Some disease-associated pathways associated with carcinogenesis, Alzheimer′ disease and Parkinson’s disease were also induced, which suggested that PAT also seemed to be associated with cancer and neuropathic disease. Taken together, our findings bring novel insight into the regulatory molecular mechanisms accounting for the toxic action of PAT against human kidney cells. 

## 5. Materials and Methods

### 5.1. Cell Culture and PAT Treatment

HEK293 cells were grown in DMEM medium supplemented with 10% fetal bovine serum (FBS), 100 U/mL of penicillin, and 100 µg/mL of streptomycin. Cells were cultured at 37 ℃, in 5% CO_2_ and 95% saturated atmospheric humidity incubator. For treatment, PAT was dissolved in sterile water to make a stocking solution of 10 mM and then added to the serumfree medium in concentrations of 2.5, 5, 7.5 and 10 μM, and these were then incubated for 10 h. The control cells were treated with serum-free medium without PAT. The 5 μM PAT dose was selected for DGE and iTRAQ analysis. For caspase 3 specific inhibitor treatment, cells were treated with 5 μM Z-DEVD-FMK (BioVision, PaloAlto, Santa Clara, CA, USA) with or without PAT. 

### 5.2. Measuring Cell Viability, LDH Leakage and Caspase Activities

2-(2-Mehtoxy-4-nitrophenyl)-3-(4-nitrophenyl)-5-(2,4-disulfophenyl)-2H-tetrazolium sodium salt (WST-8) is superior to methylthiazolyldiphenyl-tetrazolium bromide (MTT) for analyzing cell toxicity because dehydrogenase reduces WST-8 to soluble formazan in mitochondria, and WST-8 has little toxicity to cells [30]. The cell viability was determined using the Cell Counting Kit-8 (Beyotime, Beijing, China) according to the manufacturer’s instructions. LDH leakage was determined using corresponding LDH cytoxicity assay kit (Beyotime). The activities of caspase3/7 and caspase 9 were measured using corresponding colorimetric caspase-activity-assay kits (Beyotime). Both caspase 7 and caspase 3 appeared the DEVD-dependent protease activity, but caspase 3 shows a much higher efficiency in cleavaging Ac-DEVE-pNA.

### 5.3. Digital Gene Expression (DGE)

The main reagents and instruments used for deep-sequencing profiles were the Gene Expression Sample Prep Kit (Illumina, San Diego, CA, USA), the Digital Gene Expression Tag Profiling Kit (Illumina), the Solexa Sequencing Chip (Illumina), the Cluster Station and the Illumina HiSeqTM 2000 System (Illumina). Genes were deemed to be significantly differentially expressed with FDR < 0.05 and absolute value of |log2 ratio| ≥ 1 (twofold difference). 

### 5.4. iTRAQ Protein Profiling

The flow chart of the proteomic expression of 5 μM PAT on HEK293 cells is shown in Appendix A. Cells were rinsed with PBS three times and suspended in the lysis buffer (7 M urea, 2 M thiourea, 4% 3-((3-cholamidopropyl) dimethylammonium)-1-propanesulfonate (CHAPS), 40 mM Tris-HCl, pH 8.5, 1 mM phynylmethanesulfonyl fluoride(PMSF), 2 mM theylene diamine tetraacetic acid (EDTA) and sonicated in ice. The proteins were reduced with 10 mM dithiothreitol (DTT) at 56 °C for 1 h and then alkylated by 55 mM iodoacetamide (IAM) (final concentration) in the dark for 1 h. The reduced and alkylated protein mixtures were precipitated by adding 4× volume of chilled acetone at −20 °C overnight. After centrifugation at 4 °C, an aliquot of supernatant was taken for determination of protein concentration by Bradford. The protein in the supernatant were kept at −80 °C for further analysis. The detailed procedure including ITRAQ labeling, strong cation exchange (SCX) fractionation, liquid chromatography electrospray ionization tandem mass spectrometry (LC-ESI-MS/MS) analysis based on Q exactive, protein identification and quantitation. Proteins were deemed to be significantly differentially expressed with *P* values < 0.05 and fold changes of >1.2. 

### 5.5. Reverse Transcript and Real-Time Quantitative PCR Analysis of mRNA

Isolated mRNAs were reverse transcribed by using SuperRTcDNA Kit (CWBIO, Beijing, China). Quantitative real-time PCR analyses on the mRNAs were performed by using the SYBR Green Real-time PCR Master Mix (CWBIO) in a Bio-Rad IQTM5 Multicolor Real-Time PCR Detection System (Bio-Rad, Hercules, CA, USA). The thermal cycling program was set as follows: 95 °C for 10 min, followed by 40 cycles of 95 °C for 15 s and annealing/elongation at 60 °C for 1 min. The mRNA levels of target genes were normalized to the β-actin mRNA levels. The primers used in this were synthesized by Invitrogen (Shanghai, China) and the sequences were showed in Appendix A. 

### 5.6. Western Blot Analysis of Protein Expression

Cell lysates were prepared using cell lysis buffer for Western and immunoprecipitation (Beyotime) supplemented with 1 mM PMSF. Antidodies against Cyt c and β-actin were obtained from Proteintech (Wuhan, China). And antibody against HSPB8 (PA5-21612) was purchased from Invitrogen. The immunoreactive signals were developed using the ECL Western blotting Substrate (ThermoFisher, Waltham, MA, USA) and visualized on Omat X-ray films (Kodak, Rochester, NY, USA). Densitometry analysis of the images obtained from X-ray films was performed using the Image J software (v1.8.0, National Institutes of Health, Bethesda, MD, USA).

### 5.7. Statistical Analysis

Microsoft Excel 2007 and SPSS 14.0 were used for the statistical analyses. Data were expressed as mean ± standard deviation (SD). Statistical significance was evaluated by Student’s t-test. When more than one group was compared with one control, significance was evaluated according to one-way analysis of variance (ANOVA), and Duncan’s multiple-range test was applied to identify group differences. *p* < 0.05 was considered statistically significant. 

## Figures and Tables

**Figure 1 toxins-12-00681-f001:**
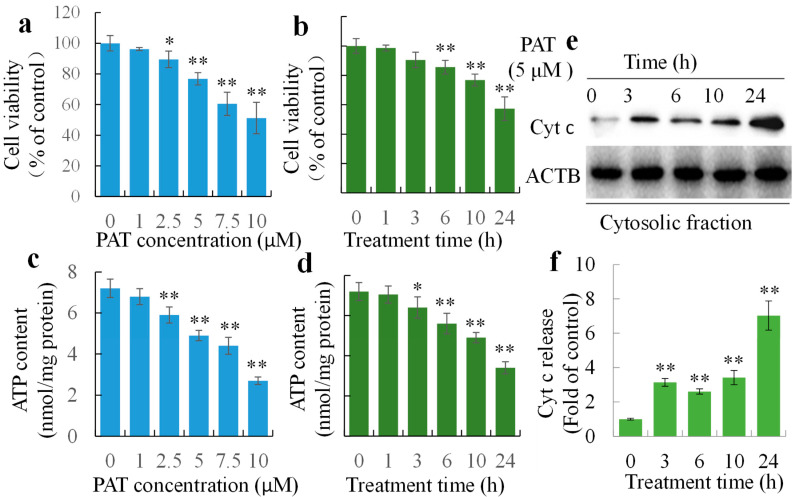
Induction of mitochondrial pathway cell death by patulin (PAT). (**a**,**b**) Cell viability reduced by PAT in dose- and time-dependent manners. (**c**,**d**) PAT induced LDH release. (**e**,**f**) Cyt c release from mitochondria to cytosol after 5 μM PAT exposure. * *p* < 0.05, ** *p* < 0.01 versus control.

**Figure 2 toxins-12-00681-f002:**
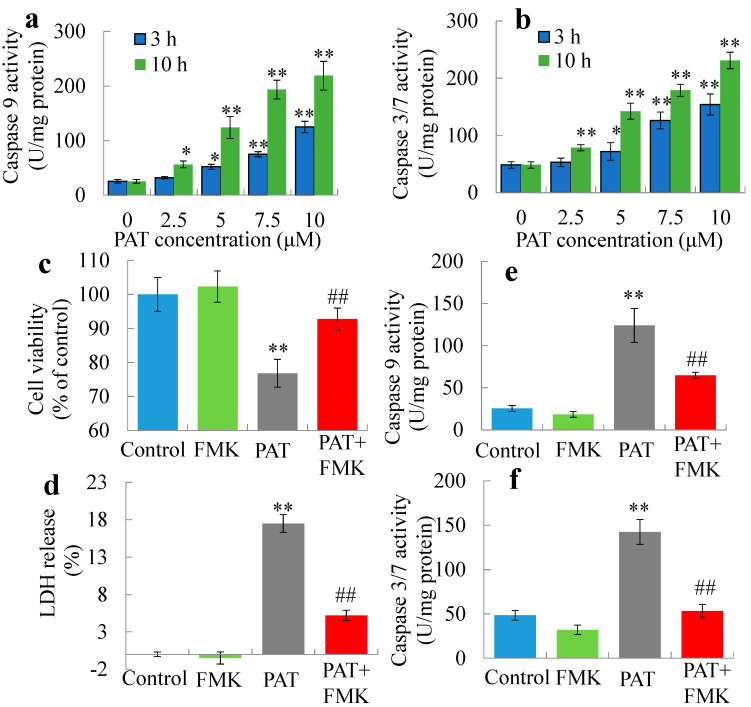
Caspase cascade pathway is involved in PAT nephrotoxicity. (**a**,**b**) The activities of initiator caspase (9) and the downstream executioner caspase (3, 7) were elevated. (**c**,**d**) Caspase 3 inhibitor Z-DEVD-FMK protected cells from PAT triggered cell damage based on cell viability and LDH leakage. (**e**,**f**) Z-DEVD-FMK almost completely reversed the increase of enzyme activities of capase 9 and 3/7 at 10 h. * *p* < 0.05, ** *p* < 0.01 versus control. ## *p* < 0.01 versus 5 μM PAT.

**Figure 3 toxins-12-00681-f003:**
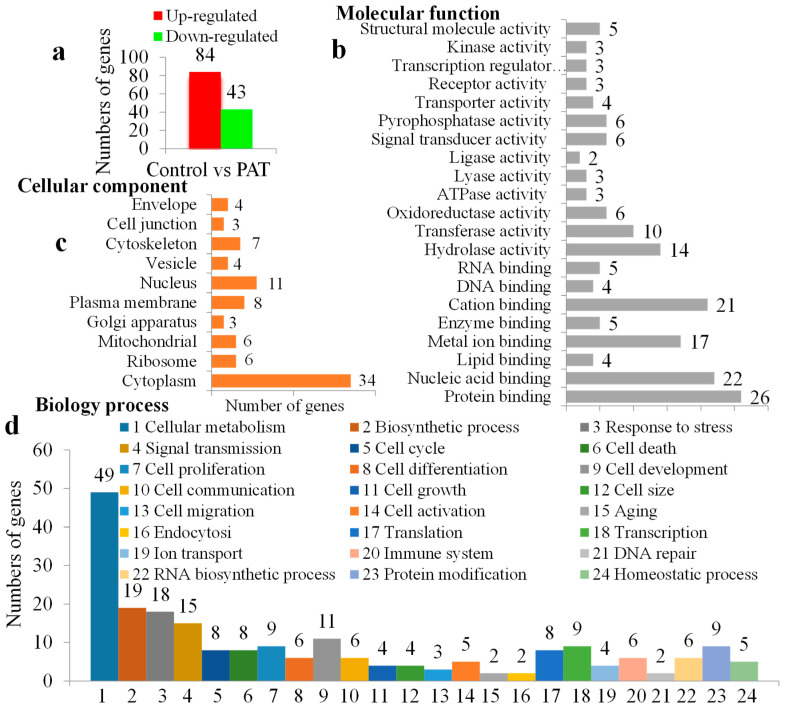
The Gene Ontology (GO) analysis of the differentially expressed genes. (**a**) Number of differentially expressed genes. (**b**–**d**) The assignment of those genes classified by GO functional categories: cell component, molecular function and biology processing separately.

**Figure 4 toxins-12-00681-f004:**
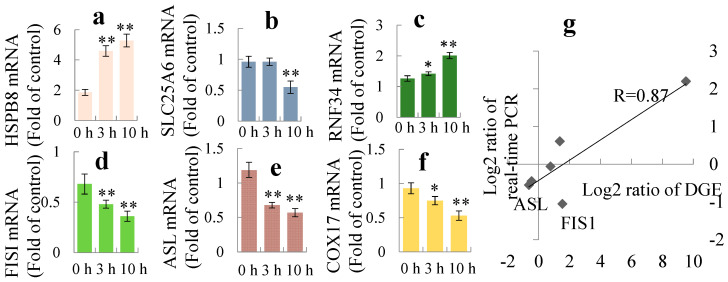
The real-time PCR validation of eight selected genes in HEK293 cells treated by 5 μM PAT at 3 h and 10 h. (**a**) Heat shock protein beta-8 (HSPB8). (**b**) Solute carrier family 25 member 6 (SLC25A6). (**c**) Ring finger protein 34 (RNF34). (**d**) Fission, mitochondrial 1 (FIS1). (**e**) Argininosuccinatelyase (ASL). (**f**) Cytochrome c oxidase copper chaperone (COX17). Beta-actin (ACTB) was used as an internal control. Each bar represents the mean of three independent experiments. * *p* < 0.05, ** *p* < 0.01 versus control. (**g**) Relationship between the gene expression measured by DGE and real-time PCR, R is the Pearson correlation coefficient.

**Figure 5 toxins-12-00681-f005:**
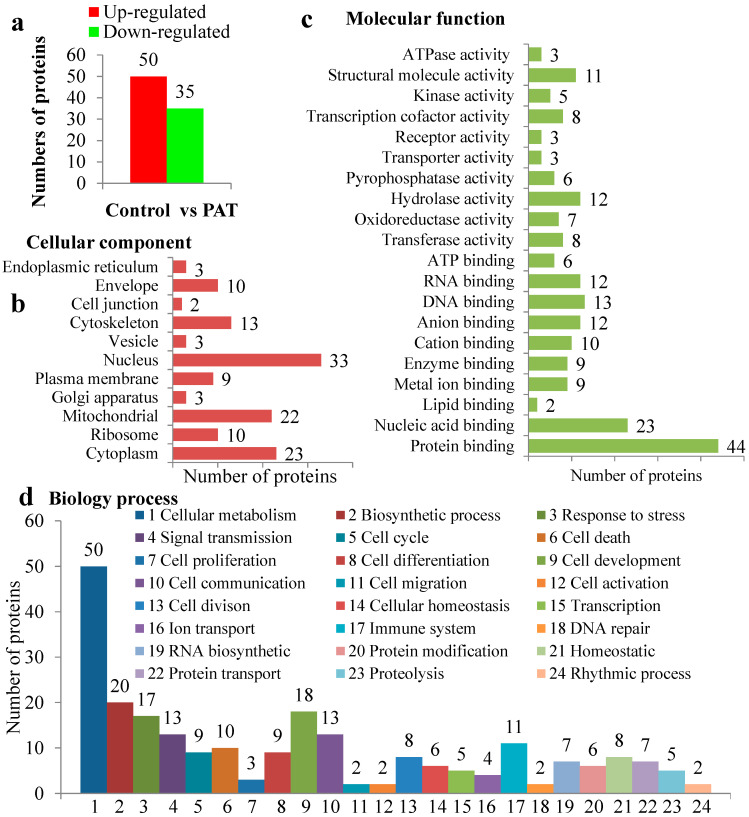
The Gene Ontology (GO) analysis of the differentially expressed genes. (**a**) The number of differentially expressed genes. (**b**–**d**) The assignment of those genes classified by GO functional categories: cell component, molecular function and biology processing separately.

**Figure 6 toxins-12-00681-f006:**
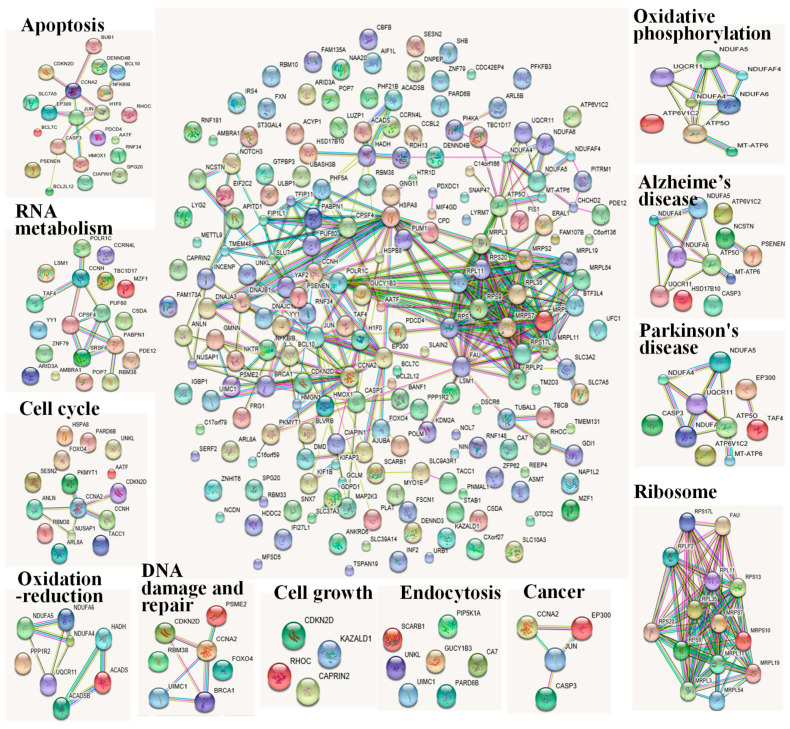
The protein–protein interaction networks of the altered gene and protein expression according to the String database. The gene names of corresponding proteins are displayed in the networks.

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
