# Peer review of "Transcriptomic and Proteomic Analysis Reveals Mechanisms of Patulin-Induced Cell Toxicity in Human Embryonic Kidney Cells"

_toxins, 2020, doi:10.3390/toxins12110681_

Round 1
Reviewer 1 Report
In this work authors investigated the novel aspects of possible mechanism of PAT cytotoxicity according to the transcriptome and proteome profiles using Digital Gene Expression (DGE) and iTRAQ proteomic approaches. The work is interesting with considerable addition to the knowledge in the field and surely will earn wide interest due to its importance. The references are up to date and comprehensive. The work is very nice and carefully discussed justifying publication in TOXINS. I suggest publication after considering the following minor remarks:
In such investigations is a key how the toxin pass the membrane. Authors mentioned, that PAT is hydrophilic, but the phospholipid bilayer plasma membrane as well as the organelle membrane is hydrophobic. Well, but the key is which ions or molecules surrounding the entering PAT. Authors mentioned that the „improved expression of endocytosis related genes and proteins indicated that this active transport form, endocytosis, might be a general pathway for PAT to entry into the cells, thus triggers the cytotoxic effects, and finally induce cell death.” I suggest to include three minor remarks according to: i) How the temperature could affect the investigated processes in the appropriate narrow temperature range of normal and fevered bodies, ii) how the temperature could affect the membrane transport of PAT and iii) how the ionic strength affects the weak interactions of PAT with the phospholipids?
Minor typos:
line 15 : induced (instead of induce)
Author Response
Dear Editors and Reviewers:
We greatly appreciate the time and effort taken by you and the reviewers to revise our manuscript. We have carefully considered your comments as well as those offered by the reviewers, and incorporated all of these helpful comments in the revised manuscript. The responses/changes are highlighted in blue in the revised manuscript.
We hope that the reviewed version can meet the journal publication requirements. We again appreciate the opportunity you gave us to revise our work. Should you have any further questions, please contact us without hesitate.
Response to comments from Reviewer 1:
In this work authors investigated the novel aspects of possible mechanism of PAT cytotoxicity according to the transcriptome and proteome profiles using Digital Gene Expression (DGE) and iTRAQ proteomic approaches. The work is interesting with considerable addition to the knowledge in the field and surely will earn wide interest due to its importance. The references are up to date and comprehensive. The work is very nice and carefully discussed justifying publication in TOXINS.
Response: Thanks. Thanks very much for your kind comments.
I suggest publication after considering the following minor remarks:
In such investigations is a key how the toxin pass the membrane. Authors mentioned, that PAT is hydrophilic, but the phospholipid bilayer plasma membrane as well as the organelle membrane is hydrophobic. Well, but the key is which ions or molecules surrounding the entering PAT. Authors mentioned that the improved expression of endocytosis related genes and proteins indicated that this active transport form, endocytosis, might be a general pathway for PAT to entry into the cells, thus triggers the cytotoxic effects, and finally induce cell death.” I suggest to include three minor remarks according to: i) How the temperature could affect the investigated processes in the appropriate narrow temperature range of normal and fevered bodies, ii) how the temperature could affect the membrane transport of PAT and iii) how the ionic strength affects the weak interactions of PAT with the phospholipids?
Response: Thank you for your good question. You provide a novel aspect to consider the toxicity mechanism of PAT in kidney cells, which we have not taken into consideration carefully before. As we know, during food processing, mycotoxins are highly stable, though different factors can affect the mycotoxin stability. Heat treatment showed a limited effect on PAT reduction in apple juice as only 26% of PAT was reduced at 100℃for 20 min. Pasteurization is the heat treatment used in fruit and vegetable juice and the reduction of PAT in apple juice in industrial conditions can not be very big and only a low PAT reduction can be expected pasteurization.
In our experiment mode, the cells were incubated at 37℃ to test the toxic effect of PAT toward kidney cell. As for how the temperature could affect the transmembrane process of PAT into the cells during the normal and fevered bodies, and how the temperature could affect the membrane transport of PAT, there were not a lot of information about those. We can speculate form the nature of PAT that the body temperature might have little influence on the reduction of PAT. But maybe the temperature should have impacts on the expression of the endocytosis as well as active transport of the exogenous substance. As for how the ionic strength affects the weak interactions of PAT with the phospholipids, we tempt go look up literatures, and found that there was little information about such aspects.
You provide new thoughts to do toxic research for us. In our further research work, we will pay attention to such thoughts. Thanks for your suggestion again.
Minor typos:
line 15 : induced (instead of induce)
Response: Thanks. We are so careless. And the word “reduce” in changed into “induced”.
Reviewer 2 Report
Review of the manuscript entitled Transcriptomic and proteomic analysis reveals mechanisms of food contaminant patulin-induced cell toxicity in human embryonic kidney cells (ID: toxins- 952945)
General comments:
The manuscript represents novel data on mechanism of PAT cytotoxicity and analysis of the transcriptome and proteome profiles in human embryonic kidney (HEK293) cells upon concentration and time dependent exposure to PAT. Results showed that significant changes involved apoptosis, oxidative phosphorylation ribosome, cell cycle, nucleotide metabolism, cell growth and endocytosis. Also, serval disease-associated pathways associated with carcinogenesis, Alzheimer' disease and Parkinson's disease were induce by PAT, showing that PAT might be associated with cancer and neuropathic disease. The manuscript is worth of publishing, but although I am not English native speaker, the English is poor and that have a negative impact on manuscript. Also, I don’t feel as expert in transcriptomics and proteomics. Thus I have no comments on methodology and presentations of results in figures 2-4.
Specific comments
Title is too long, “food contaminant” can be excluded. I suggest changing the title into Transcriptomic and proteomic analysis reveals mechanisms of patulin toxicity in human embryonic kidney cells.
Introduction
Line 25: Aspergillus and Penicillium species (genera should be in italic)
Line 38: Clarify instead of clarity
Results
In description of Figure 1 is Q-VD-OPH inhibitor but in Fig 1 as well as in the text of chapter 2.1. is Z-DEVD-FMK, please clarify this. Also, figure 1 (ijkl) is confusing, what is on abscissa? In my opinion it is more appropriate to have all parameters in abscissa of all graphs. Figure 1 is missing the legend for *, ** and ## -
Figure 2. What is on abscissa in graphs e-j?
Lines 155-158 as well as 161-162 should be in discussion
Discussion
Line 104: Rewrite the first sentence – English revision!
Line 205: brackets are empty?
Conclusions
I suggest to delete “in summary”. Here is suggestion for conclusions:
A comprehensive transcriptome and proteomics analyses that combined two unbiased and high throughput techniques were applied for the first time to define the mechanism of PAT cytotoxicity. Genes and proteins related to apoptosis, oxidative phosphorylation ribosome, cell cycle, nucleotide metabolism, cell growth and endocytosis were differentially expressed in human embryonic kidney cells HEK293 cells upon exposure to PAT. In addition, some pathways associated with carcinogenesis, Alzheimer' disease and Parkinson's disease were also induced, which suggests that PAT intake may result in cancer and/or neuropathic disease.
Materials and methods
What was the control in cell treatment? Water?
Author Response
Dear Editors and Reviewers:
We greatly appreciate the time and effort taken by you and the reviewers to revise our manuscript. We have carefully considered your comments as well as those offered by the reviewers, and incorporated all of these helpful comments in the revised manuscript. The responses/changes are highlighted in blue in the revised manuscript.
We hope that the reviewed version can meet the journal publication requirements. We again appreciate the opportunity you gave us to revise our work. Should you have any further questions, please contact us without hesitate.
Response to comments from Reviewer 2:
General comments:
The manuscript represents novel data on mechanism of PAT cytotoxicity and analysis of the transcriptome and proteome profiles in human embryonic kidney (HEK293) cells upon concentration and time dependent exposure to PAT. Results showed that significant changes involved apoptosis, oxidative phosphorylation ribosome, cell cycle, nucleotide metabolism, cell growth and endocytosis. Also, serval disease-associated pathways associated with carcinogenesis, Alzheimer' disease and Parkinson's disease were induce by PAT, showing that PAT might be associated with cancer and neuropathic disease. The manuscript is worth of publishing, but although I am not English native speaker, the English is poor and that have a negative impact on manuscript. Also, I don’t feel as expert in transcriptomics and proteomics. Thus I have no comments on methodology and presentations of results in figures 2-4.
Response:
Specific comments
Title is too long, “food contaminant” can be excluded. I suggest changing the title into Transcriptomic and proteomic analysis reveals mechanisms of patulin toxicity in human embryonic kidney cells.
Response: Thanks for your kind suggestion. It is really quite concise to delete “food contaminant” yet doesn’t affect the overall meaning. And we have adopted your suggestion.
Introduction
Line 25: Aspergillus and Penicillium species (genera should be in italic)
Response: Thanks. Thanks for your good recommendation. We have revised it in the manuscript.
Line 38: Clarify instead of clarity
Response: Thanks for your kind reminding. The word “clarity” is replaced by clarify in line 41.
Results
In description of Figure 1 is Q-VD-OPH inhibitor but in Fig 1 as well as in the text of chapter 2.1. is Z-DEVD-FMK, please clarify this.
Response: Thanks again for your reminding. Here the inhibitor in the description of Figure 1 should be Z-DEVD-FMK, and we have revised in the description of figure 1. Q-VD-OPH is a pancaspase inhibitor as Z-DEVE-FMK is caspase 3 specific inhibitor. Although we have bought both reagents to test the action, here shows the results of capase 3 inhibitor.
Also, figure 1 (ijkl) is confusing, what is on abscissa? In my opinion it is more appropriate to have all parameters in abscissa of all graphs.
Response: Thanks very much. After careful consideration, we have redrawn part of the histogram and separate figure 12 figures to make it more clearly. And the finally figures are attached.
Figure 1 is missing the legend for *, ** and ## -
Response: Thanks for your suggestion. “* P<0.05, **P<0.01 versus control. ##P<0.01 versus 5 μM PAT” is added in the last in description of figure 1.
Figure 2. What is on abscissa in graphs e-j?
Response: Thanks for your suggestion. We are sorry for the missing of the marks of abscissa when we assemble the figures. And the abscissa represent the hours (3, 10, 24) after treatment is added in figure 2.
Lines 155-158 as well as 161-162 should be in discussion
Response: Thanks for your good suggestion. The sentence “PAT is hydrophilic, but the phospholipid bilayer plasma membrane as well as the organelle membrane is hydrophobic. The improved expression of endocytosis related genes and proteins indicated that this active transport form, endocytosis, might be a general pathway for PAT to entry into the cells, thus triggers the cytotoxic effects, and finally induce cell death.” is move to the discussion part form line --to line --.
Discussion
Line 104: Rewrite the first sentence – English revision!
Response: Thank you very much. The sentence “To obtain more information about the nature of PAT-induced toxicity, an iTRAQ-based approach was pursued to quantify the global protein expression profiles.” was changed into “To obtain more information about the nature of PAT-induced toxicity, an iTRAQ-based approach was pursued to quantify the global protein expression profiles”.
Line 205: brackets are empty?
Response: Thank you very much. The reference number [29] was added in the bracket.
Conclusions
I suggest to delete “in summary”.
Here is suggestion for conclusions:
A comprehensive transcriptome and proteomics analyses that combined two unbiased and high throughput techniques were applied for the first time to define the mechanism of PAT cytotoxicity. Genes and proteins related to apoptosis, oxidative phosphorylation ribosome, cell cycle, nucleotide metabolism, cell growth and endocytosis were differentially expressed in human embryonic kidney cells HEK293 cells upon exposure to PAT. In addition, some pathways associated with carcinogenesis, Alzheimer' disease and Parkinson's disease were also induced, which suggests that PAT intake may result in cancer and/or neuropathic disease.
Response: Thanks. We have delete the words “in summary” at the beginning of the conclusion part.
Materials and methods
What was the control in cell treatment? Water?
Response: Thanks. As patulin is soluble in water, so PAT is dissolved in sterile water to make a stock solution of 10 mM and further diluted to final concentrations of 2.5, 5, 7.5 and 10 μM with serum-free medium. So the control cells is treated with serum-free medium without PAT. And we added the control sentence to make it more clearly in line –“The control cells were treated with serum-free medium without PAT”.

Reviewer 3 Report
The publication presents very important elements and an interesting comprehensive interpretation of interdisciplinary issues.
The reviewer suggests a change in the interpretation of the results. The graphs in Figures 1-3 are unreadable and too many of them. The reviewer suggests to improve the graphics so that the results are legible. The description of the results is also illegible. the reviewer suggests using generalities when interpreting the results, as describing the results that can be read from the graph is unnecessary. Please edit the description of the results.
Figure 4 is worth emphasizing, which is very nicely described in the text. Please arrange the conclusions. Most of the information in this chapter is observations.
Author Response
Dear Editors and Reviewers:
We greatly appreciate the time and effort taken by you and the reviewers to revise our manuscript. We have carefully considered your comments as well as those offered by the reviewers, and incorporated all of these helpful comments in the revised manuscript. The responses/changes are highlighted in blue in the revised manuscript.
We hope that the reviewed version can meet the journal publication requirements. We again appreciate the opportunity you gave us to revise our work. Should you have any further questions, please contact us without hesitate.
Response to comments from Reviewer 3:
The publication presents very important elements and an interesting comprehensive interpretation of interdisciplinary issues.
The reviewer suggests a change in the interpretation of the results. The graphs in Figures 1-3 are unreadable and too many of them. The reviewer suggests to improve the graphics so that the results are legible.
Response: Thanks very much for your kind suggestion. After careful consideration, we have redrawn part of the histogram and separate figure 1 and figure 2 into 4 figures. And the finally figures are attached.
The description of the results is also illegible. the reviewer suggests using generalities when interpreting the results, as describing the results that can be read from the graph is unnecessary. Please edit the description of the results.
Response: Thanks very much for your suggestion. We have checked the description of the results carefully, and rewrite the results as much as possible. And the changed parts were marked in blue.
Figure 4 is worth emphasizing, which is very nicely described in the text. Please arrange the conclusions. Most of the information in this chapter is observations.
Response: Thanks for your kind suggestion. We have rearrange the conclusions and the final conclusion is as follows:
“This study presents that PAT triggers caspase-dependent cell death via the intrinsic apoptotic pathway. A comprehensive transcriptome and proteomics analyses that combined two unbiased and high throughput techniques for the first time to define the mechanism of the nephrotoxicity of PAT. Genes and proteins related to apoptosis, oxidative phosphorylation ribosome, cell cycle, nucleotide metabolism, cell growth and endocytosis were differentially expressed in HEK293 cells induced by PAT. Especially, caspase 3 appeared to be particularly crucial in PAT induced cell apoptosis and the elevated expression of UQCR11 might be able to explain the increase of ROS. In addition, the activation active transport form and endocytosis, might be a general way for PAT to entry into the cells. Some disease-associated pathways associated with carcinogenesis, alzheimer' disease and parkinson's disease were also induced, which suggested that PAT also seemed to be associated with cancer and neuropathic disease. Taken together, our findings bring novel insight into the regulatory molecular mechanisms accounting for the toxic action of PAT against human kidney cells. ”
Also we also rearrange the abstract part. The final abstract is shown as follows:
“Patulin (PAT) is a natural mycotoxin that commonly contaminates fruits and fruit-based products. Previous work indicated that PAT induced apoptosis in which reactive oxygen species (ROS) is involved in human embryonic kidney (HEK293) cells. To uncover novel aspects of possible mechanism of PAT nephrotoxicity, the transcriptome and proteome profiles were investigated using Digital Gene Expression (DGE) and iTRAQ proteomic approaches. A total of 127 genes and 85 proteins were found to express differentially in response to 5 μM PAT for 10 h in HEK293 cells. The most dramatic changes of expression were noticed with genes or proteins related to apoptosis, oxidative phosphorylation ribosome, and cell cycle. Especially, the activation of caspase 3, UQCR11, active transport form and endocytosis appeared to be crucial in PAT kidney cytotoxicity. PAT also seemed to be associated with cancer and neuropathic disease as pathways associated with carcinogenesis, alzheimer' disease and parkinson's disease were activated. Overall, this study served to uncover overall insights associated with signaling pathway that modulated PAT toxicity mechanism.”
If you have any more suggestions, please feel free to tell us. Thank you!

Round 2
Reviewer 2 Report
The manuscript has been improved and I recommend its publication